# Harnessing the β-boron effect for regioselective Ru-catalyzed hydrosilylation of internal alkynes

Jiasheng Qian[1,4], Shuang Lin[1,4], Zhi-Hao Chen[1], Jiawu Huang[1], Wenjin Zhang[1], Qingjiang Li ®[1], Tian-Yu Sun ®[2] ✉ & Honggen Wang ®[1,3] ✉

Metal-catalyzed hydrosilylation of alkynes is recognized as a straightforward and atom economic method for synthesizing alkenylsilanes. While substantial advancements have been made with terminal alkynes, achieving precise regio- and stereocontrol with unsymmetrical internal alkynes remains a significant challenge. In this study, we report the utilization of an intriguing β-boron effect in metal catalysis, enabling an exclusively regioselective Ru-catalyzed hydro-silylation of propargylic N-methyliminodiacetic acid boronates (B(MIDA)) to synthesize alkenylsilanes. Variations in the Ru catalyst can lead to stereo-divergency without compromising regioselectivity. Density functional theory (DFT) calculations indicate that the hyperconjugative effect of the σ(C−B) bond, which stabilizes the electrophilic metallacyclopropene intermediate with Fischer carbene character, is crucial for achieving high regioselectivity. The observed switch in stereoselectivity is attributed to the different steric effects of 1,2,3,4,5-pentamethylcyclopenta-1,3-diene (Cp*) and cyclopenta-1,3-diene (Cp) ligands in the catalyst. This method produces a diverse array of regio- and stereodefined products incorporating boryl, silyl, and alkene functionalities, each of which serves as a valuable handle for further functionalization.

Organosilicon compounds hold significant utility across diverse fields including synthetic chemistry[1–7], medicinal chemistry[8], and materials science[9]. Among these, alkenylsilanes stand out as crucial building blocks in organic synthesis. Their possession of a polarized alkene moiety and a silicon center renders them environmentally friendly and user-friendly carbon-centered nucleophiles, facilitating numerous nucleophilic addition[10,11], oxidation[12–14], and cross-coupling reactions[2,3].

While numerous methods exist for synthesizing alkenylsilanes, metal-catalyzed hydrosilylation of alkynes offer the most direct and atom-economic approach[15,16]. However, the primary focus has been on the hydrosilylation of terminal alkynes[17–26]. In contrast, the utilization

of unsymmetrical internal alkynes has faced challenges in achieving precise regio- and stereocontrol (Fig. 1A). Notably, employing unsymmetrical (internal) alkynes can yield up to four regio- and/or stereoisomers. To tackle this issue, attaching electronically (e.g., aryl[24,27,28], carbonyl[29–32], CF₃[33], or heteroatom[34–37]) or sterically diverse substituents[38–41] directly to the acetylenic carbon has proven effective (Fig. 1B). In addition, for internal alkynes with similar substituents, utilizing pendent directing groups has emerged as another efficient protocol[14,42–46]. Specifically, directing groups based on electron-negative heteroatoms capable of coordinating[14,42,43,46] with the metal center or hydrogen-bonding[44–47] to the metal ligand have been

[1]State Key Laboratory of Anti-Infective Drug Discovery and Development, Guangdong Provincial Key Laboratory of Chiral Molecule and Drug Discovery, School of Pharmaceutical Sciences, Sun Yat-sen University, Guangzhou, China. [2]Key Laboratory of Computational Chemistry and Drug Design, State Key Laboratory of Chemical Oncogenomics, Shenzhen Key Laboratory of Chemical Genomics, School of Chemical Biology and Biotechnology, Peking University Shenzhen Graduate School, Shenzhen, China. [3]State Key Laboratory of Coordination Chemistry, School of Chemistry and Chemical Engineering, Nanjing University, Nanjing, China. [4]These authors contributed equally: Jiasheng Qian, Shuang Lin. ✉e-mail: Tian-Yu_Sun@pku.edu.cn; wanghg3@mail.sysu.edu.cn

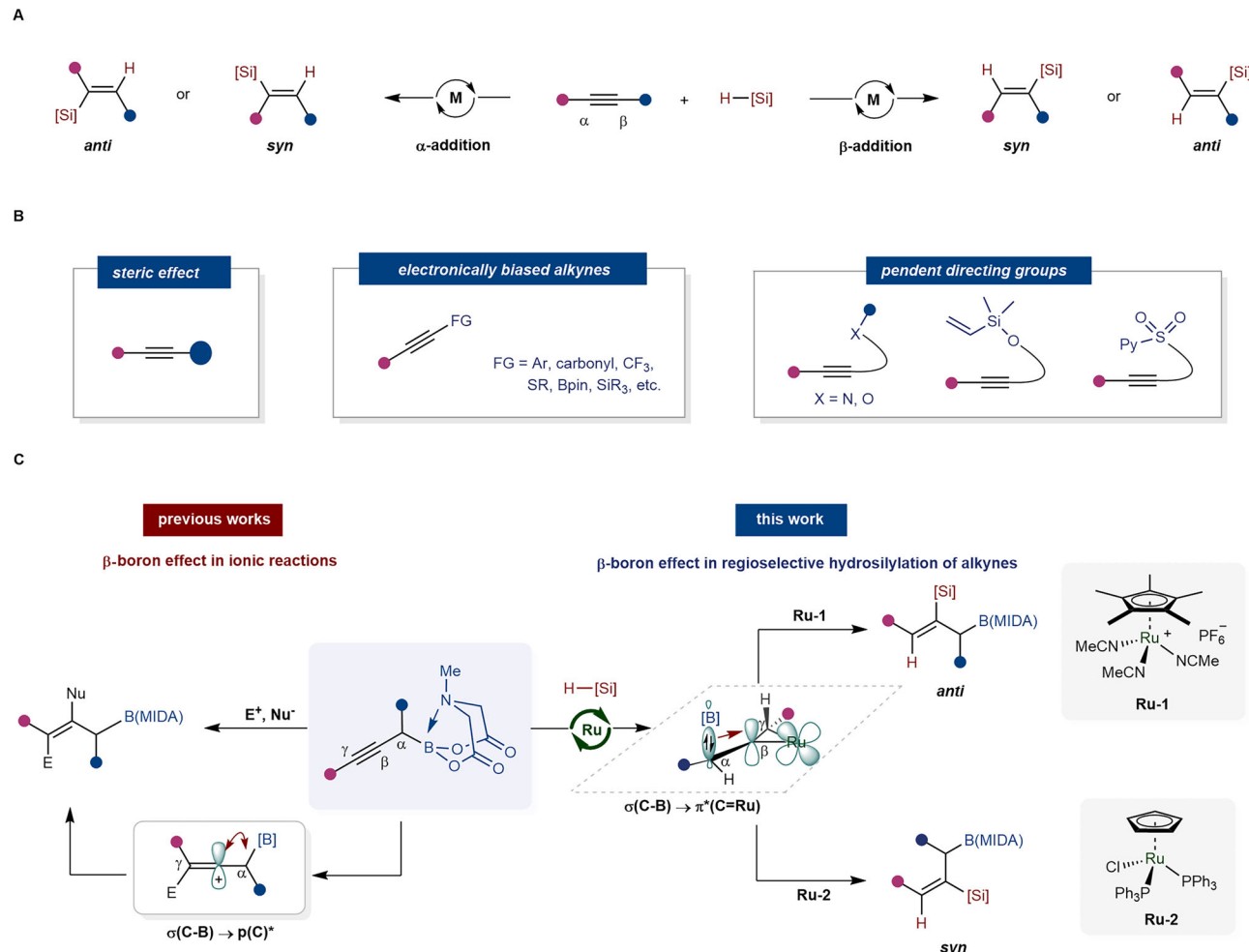

**Fig. 1 | Functional groups effect in hydrosilylations of internal alkynes. A** metal-catalyzed hydrosilylation of internal alkynes. **B** traditional strategies for solving regioselectivity problems. **C** β-boron effect-enabled regioselective functionalizations of alkynes.

elegantly developed (Fig. 1B). Despite these advancements, there remains a pressing demand for alternative strategies to achieve precise regio- and stereocontrol for practical applications.

Recently, we have unveiled the capability of the $sp^3$-B hybridized B(MIDA) (*N*-methyliminodiacetic acid) moiety[48,49] to induce the high-lying σ(C−B) orbital interaction with the low-lying p*(C), effectively stabilizing adjacent carbocations. The β-boron effect has enabled us to achieve regioselective functionalization reactions involving internal alkenes[50,51], alkynes[52], and allenes[53] (left, Fig. 1C). Motivated by this discovery, we sought to explore whether a similar effect could be harnessed in the realm of regio-controlled metal catalysis.

Our attention was drawn to the intriguing Ru-catalyzed hydro-silylation of alkynes, a transformation pioneered by Trost[19,54] and further developed by Fürstner[44,47], among others[32,35,38,45,55]. Mechanistically, an unusual oxidative hydrometallation of alkyne leads to the formation of a metallacyclopropene intermediate, wherein the hydrogen atom and the silyl group of the silane are arranged on opposite sides of the ruthenacyclopropene plane. Following a silyl migration, an unusual *anti*-addition product is formed[54–56]. We hypothesized that the electrophilic nature of the metallacyclopropene intermediate, akin to a Fischer carbene[57], could potentially benefit from stabilization by an adjacent σ(C−B) bond, thereby facilitating high regioselectivity (right, Fig. 1C). This hypothesis aligns with the proton-like property of the hydrogen atom in Ru−H[35,54] and the nucleophilic nature of the γ position of propargylic B(MIDA)s[52]. Consequently, the β-silylated products should be preferentially formed.

In this report, we present a β-boron effect-guided regioselective Ru-catalyzed hydrosilylation of propargylic B(MIDA)s. Intriguingly, subtle variations in the Ru catalyst led to a switch in stereoselectivity without altering the regioselectivity. Such a pronounced alteration in stereo-selectivity is a rare occurrence in metal-catalyzed hydrosilylation of internal alkynes. This reaction enables the synthesis of diverse regio- and stereo-defined building blocks that amalgamate the synthetic potential of boryl, silyl, and alkene functionalities (right, Fig. 1C)[58–66]. The involvement of the β-boron effect in stabilizing the electrophilic Ru-carbene is theoretically supported, accounting for the observed high level of regioselectivity. Furthermore, our findings regarding stereo-divergence have been rationalized through DFT studies.

## Results

In our initial study, we employed propargylic B(MIDA) **S-1** and trie-thoxysilane as model starting materials (Table 1). The cationic catalyst [Cp*Ru(MeCN)₃]PF₆ was chosen for initial evaluation due to its proven catalytic reactivity in a range of alkyne functionalization reactions. When conducted in THF, the reaction provided the exclusive β-selective (β/γ > 20:1) product **1**, validating our working hypothesis. Additionally, exceptional *anti*-selectivity (Z/E > 20:1) and yield (95%) were achieved (entry 1). The reaction performed equally well in 1,4-dioxane (entry 2), but MeCN proved unsuitable as a solvent, likely due to its strong coordinating nature (entry 3)[67]. An intriguing observation arose when we substituted [Cp*Ru(MeCN)₃]PF₆ with its analogue [CpRu(MeCN)₃]PF₆, featuring a smaller Cp ligand. This change resulted

**Table 1 | Condition Optimization**

| entry | catalyst | yield[a] | solvent | time | selectivities[b] |
|---|---|---|---|---|---|
| 1 | $[Cp^*Ru(MeCN)_3]PF_6$ | 95% | THF | 30 min | $Z/E > 20:1$, $\beta/\gamma > 20:1$ |
| 2 | $[Cp^*Ru(MeCN)_3]PF_6$ | 95% | dioxane | 30 min | $Z/E > 20:1$, $\beta/\gamma > 20:1$ |
| 3 | $[Cp^*Ru(MeCN)_3]PF_6$ | NR | MeCN | 12 h | - |
| 4 | $[CpRu(MeCN)_3]PF_6$ | 97% | THF | 30 min | $E/Z = 1:1$, $\beta/\gamma > 20:1$ |
| 5 | $CpRu(PPh_3)_2Cl$ | 65% | THF | 2 h | $E/Z = 16:1$, $\beta/\gamma > 20:1$ |
| 6 | $Cp^*Ru(PPh_3)_2Cl$ | 89% | THF | 2 h | $E/Z = 1:1$, $\beta/\gamma = 3:1$ |
| **7** | $CpRu(PPh_3)_2Cl$ | 84% | dioxane | 3 h | $E/Z > 20:1$, $\beta/\gamma > 20:1$ |
| 8 | $[Cp^*RuCl_2]_n$ | NR | THF | 12 h | **-** |
| 9 | $Ru(PPh_3)_3Cl_2$ | NR | dioxane | 12 h | - |
| 10 | $[Ru(pcymene)Cl_2]_2$ | NR | THF | 12 h | - |
| 11 | $[Rh(cod)_2]BF_4$ | 96% | DCM | 1 h | $E/Z > 20:1$, $\beta/\gamma = 1:1$ |
| 12 | $Co_2(CO)_8$ [c] | 83% | DCE | 3 h | $E/Z > 20:1$, $\beta/\gamma = 6:1$ |
| 13 | $Pt(PPh_3)_4$ | NR | DCM | 12 h | - |

[a]Isolated yield. [b]The ratios were determined by $^1H$ NMR spectroscopy. [c]Reaction temperature: 80 °C. MIDA: *N*-methyliminodiacetic acid; Cp*:1,2,3,4,5-pentamethylcyclopenta-1,3-diene; Cp: cyclopenta-1,3-diene.

in a switch of stereochemistry, with the *syn*-addition product increasing ($E/Z = 1:1$), while maintaining high regioselectivity (entry 4). This outcome underscores the significant impact of ligand steric effects on stereoselectivity[28,38,68,69] and emphasizes the potential for achieving stereo-divergence through judicious catalyst selection. Indeed, employing $CpRu(PPh_3)_2Cl$ as the catalyst in THF led to excellent *syn*-selectivity and regioselectivity, yielding compound **2** with a 65% yield, $E/Z = 16:1$, and $\beta/\gamma > 20:1$ (entry 5). Conversely, the use of the bulkier $Cp^*Ru(PPh_3)_2Cl$ catalyst resulted in increased formation of the *anti*-addition product (entry 6). Conducting the reaction in 1,4-dioxane improved both yield and stereoselectivity (entry 7). Exploration of other ruthenium-based catalysts such as $[Cp^*RuCl_2]_n$, $Ru(PPh_3)_3Cl_2$, and $[Ru(pcymene)Cl_2]_2$ proved ineffective (entries 8–10). In addition, attempts with other hydrosilylation catalysts, including $[Rh]$[70], $[Co]$[41], and $[Pt]$[37] complexes following either the classic Chalk-Harrod or modified Chalk-Harrod mechanism, yielded either poorer regioselectivity or no reactivity (entries 11–13).

The scope for the synthesis of both stereoisomers was then explored (Fig. 2). A diverse array of silanes, including trialkyl- (**3-5**), aryldialkyl- (**6**), mono-alkoxyl- (**7**), di-alkoxyl- (**8, 9**), and trialkoxyl-substituted (**10, 11**) ones, were successfully applied to the *anti*-addition reactions, yielding the corresponding products with excellent regio- and stereoselectivities. The lower $Z/E$ ratio (3:1) for $HSiMe(OTMS)_2$ is likely due to steric hindrance (**9**). Next, a wide variety of internal alkynes were subjected to the reaction. It was found that valuable functional groups such as halogens (**14-16**), aryl groups (**17–22**), cyano (**26**), sulfonamide (**27, 29**), amide (**28, 30**), ester (**31**), hydroxy (**32-34**), OTBDPS (**35, 36**), OBz (**37**), OTs (**38**), and OBn (**39**) were all well tolerated, demonstrating the robustness and mildness of the protocol. Terminal alkynes were applicable as well, but in lower yields, with about 50% starting materials remained (**41, 42**). The installation of a secondary alkyl substituent on the distal side of the triple bond had minimal impact on the selectivity (**40**). However, the introduction of a substituent α- to the boryl moiety did reduce the $Z/E$ ratio to 4.5:1, but the regioselectivity remained unaffected (**43**). Enyne (**44**) was also well tolerated with good yield and stereoselectivity. However, aromatic alkyne (**45**) was not applicable even when heated to 100 °C.

The scope for the $CpRu(PPh_3)_2Cl$-catalyzed *syn*-hydrosilylation was also impressive (Fig. 3). While all the silanes tested exhibited excellent regioselectivity, trialkyl- (**46, 47**) and aryldialkylsilanes (**48**) showed slightly diminished stereoselectivity. Notably, $HSi(OTMS)_2Me$ performed well in this reaction (**49**), and the resulting vinyl-$Si(OTMS)_2Me$ is a suitable Hiyama-Denmark cross-coupling partner[22]. For various propargylic B(MIDA)s, consistently high levels of regio-

control were observed. It is noteworthy that, in previous studies, alkynes containing tethered hydrogen bond donors (e.g., OH, NHTf) that can induce inter-ligand interactions with the [Ru–Cl] bond and the silanes tend to undergo proximal silylation to the directing group[44,45,47]. However, in our case, the presence of the B(MIDA) moiety reversed this regioselectivity, yielding exclusively the distal functionalization product (**61, 63, 64**). These results highlight the profound directing effect of the B(MIDA) moiety. Our DFT calculations revealed that the activation energies for compounds **59, 61,** and **62**, influenced by the directing effect of the B(MIDA) moiety, are 2.7–4.8 kcal/mol lower than those governed by hydrogen bonding (For further details, see the Supplementary Information). Propargylic B(MIDA)s with cyclic (**58, 69**) or branched substituent (**59, 68**) were also competent substrates. Commonly encountered functional groups such as chloro (**53**), aryl (**54–57**), cyano (**60**), amide (**61**), ester (**62**), hydroxy (**63, 64**), OTBDPS (**65**), OBz (**66**), and OTs (**67**) were all well tolerated. Interestingly, this protocol is sensitive to steric hindrance at the B(MIDA) side. When a methyl group was installed at its α position, increasing the reaction temperature to 60 °C was necessary to ensure high conversion, which unfortunately compromised both regio- and stereoselectivity. In this circumstance, we found that replacing $CpRu(PPh_3)_2Cl$ with its cationic analog $[CpRu(MeCN)_3]PF_6$ was beneficial (**70**). However, enyne (**71**), aromatic alkyne (**72**), and terminal alkyne (**41**) was not applicable.

## Synthetic utilities

The presence of chemically distinct boryl, silyl, and alkene functionalities in the products offers ample opportunities for follow-up decoration. As shown in Fig. 4, the gram-scale synthesis of **6** could be achieved in 87% yield with simple crystallization. The silyl group in **6** could be easily converted to iodide **73** upon treatment with NIS and 2,6-lutidine, which would allow further cross-coupling reactions such as Suzuki-Miyaura cross-coupling (**74**), Sonogashira cross-coupling (**75**), and Mizoroki-Heck cross-coupling (**76**). The B(MIDA) moiety can be transferred to Bpin, and the resulting allyl Bpin can be chemoselectively oxidized to a silylated allyl alcohol (**77**). The silyl group can undergo Hiyama-Denmark cross-coupling without difficulty (**78**). In addition, the stereochemistry of the double bond in the *syn*- or *anti*-product can be effectively translated into the high diastereoselectivity when reacting with benzaldehyde (**79–81**)[71,72].

## Experimental mechanistic studies

To highlight the role of the B(MIDA) moiety in this regio- and stereoselective reaction, several control experiments were conducted. The

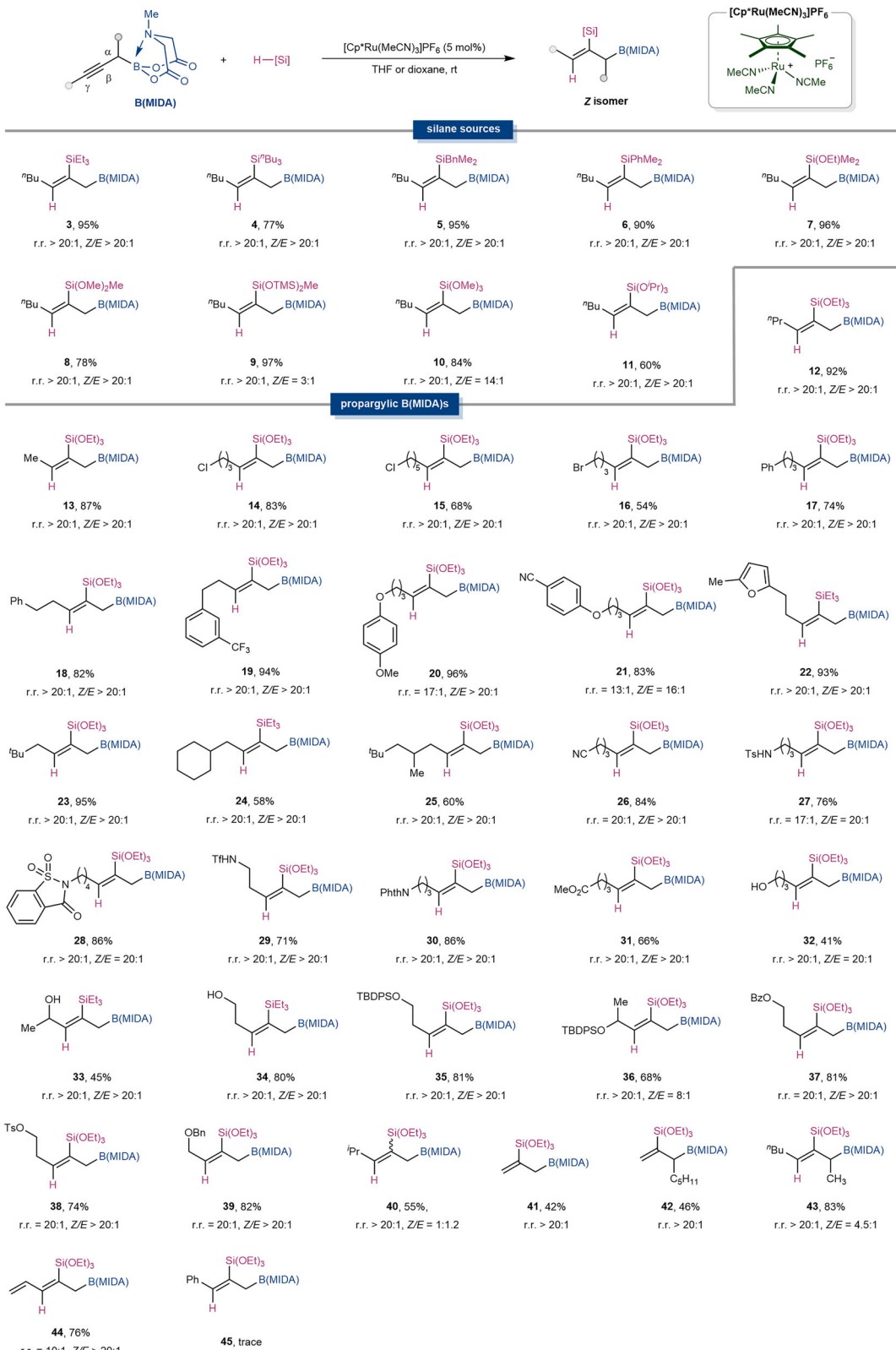

**Fig. 2 | [Cp*Ru(MeCN)₃]PF₆-catalyzed *anti*-hydrosilylation.** Reaction conditions: propargylic B(MIDA) (0.1 mmol, 1.0 equiv.), silane (0.2 mmol, 2.0 equiv.), [Cp*Ru(MeCN)₃]PF₆ (5 mol%), in 1,4-dioxane or THF (1.0 mL), room temperature.

use of sp²-B propargylic Bpin **S-40** in this reaction also led to an *anti*-addition product, but with a diminished β/γ ratio of 4:1 (Fig. 5A). Our calculations indicated that Bpin exhibits a lower tendency to engage in σ(C−B) hypercatalyzation (For further details, see the Supplementary Information). In contrast, employing propargylic TIDA (tetramethyl *N*-

methyliminodiacetic acid) boronate **S-41**, a bulkier sp³-B analogue of B(MIDA), resulted in the corresponding product with excellent regioselectivity as well. When using propargylic BF₃K **S-42**, the reaction did not occur, even when the reaction was performed in various solvents such as THF, dioxane, MeCN, or acetone to rule out solubility

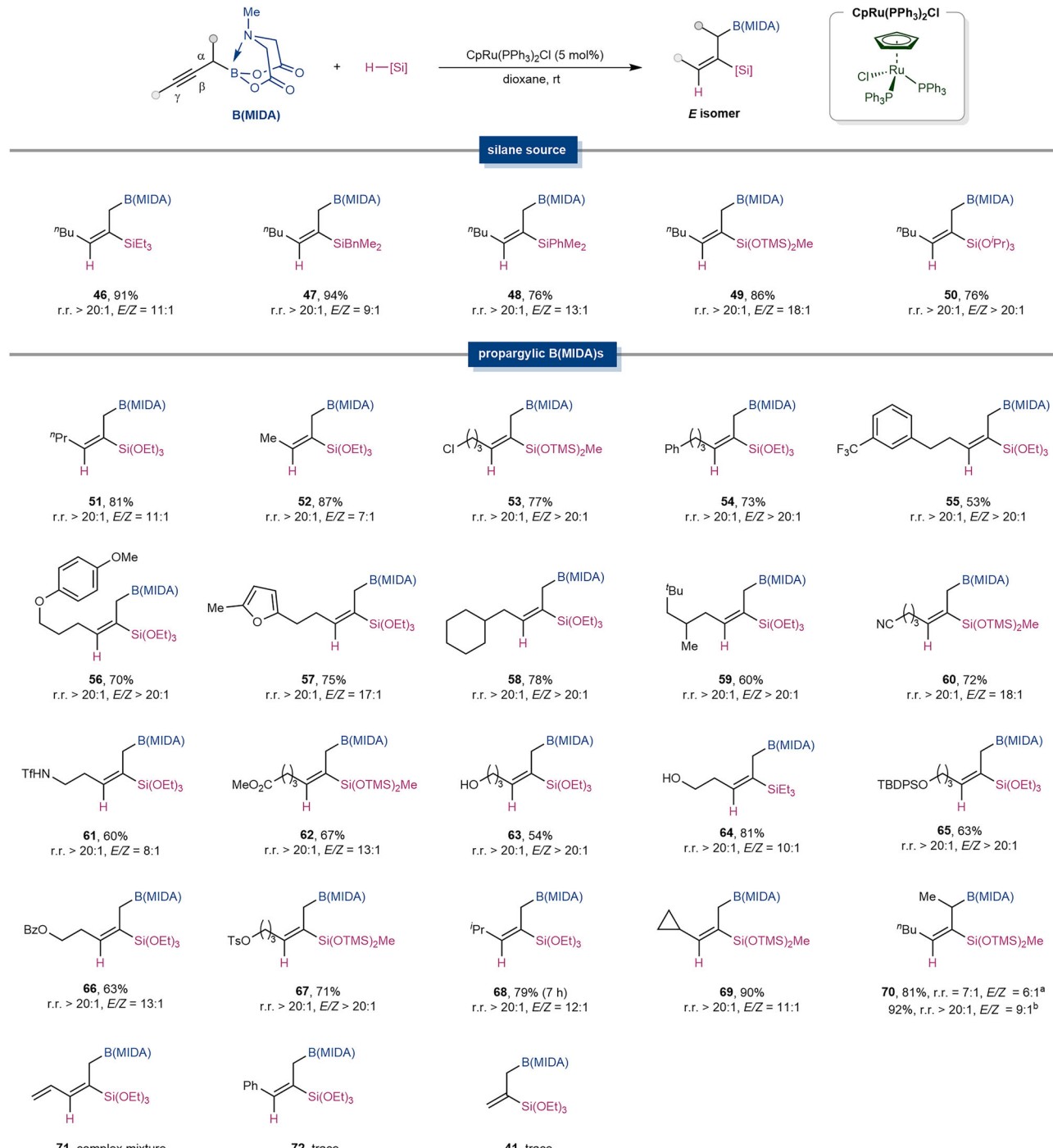

**Fig. 3 | CpRu(PPh₃)₂Cl-catalyzed *syn*-hydrosilylation.** Reaction conditions: propargylic B(MIDA) (0.1 mmol, 1.0 equiv.), silane (0.2 mmol, 2.0 equiv.), CpRu(PPh₃)₂Cl (5 mol%), in 1,4-dioxane (1.0 mL) at room temperature. [a]Reaction temperature: 60 °C, [b][CpRu(MeCN)₃]PF₆ (5 mol%) as the catalyst.

issues. Bdan **S-43** also demonstrated excellent β-selectivity, though with a relatively low yield. Taken together, these results suggest that the β-boron effect may be present in multiple boron species, but B(MIDA) delivers the most favorable outcomes. The reaction of homopropargylic MIDA-boronate **S-44** with one additional carbon linkage showed a regioselectivity of 4:1 (Fig. 5B). This result indicates that the directing ability of the B(MIDA) moiety decreases with increased distance. The hydrosilylation of hex-4-yn-1-ylbenzene, an internal alkyne without boron groups, showed a regioselectivity of 2:1 (**87, 88**). It should also be mentioned that, when using CpRu(PPh₃)₂Cl as the catalyst, the hydrosilylation of hex-4-yn-1-

ylbenzene showed both low regio- and stereoselectivity, forming including four isomers (*Z*-α, *Z*-β, *E*-α, *E*-β) (Fig. 5C). We were also curious about the origin of the unusual *syn*-selectivity observed with CpRu(PPh₃)₂Cl as the catalyst. When 5 mol% of AgPF₆ was added to precipitate the chloride from the catalyst, we observed a complete loss of stereoselectivity. When 10 mol% of PPh₃ was added to the reaction with [CpRu(MeCN)₃]PF₆ as the catalyst, the reaction did not occur. When both PPh₃ and LiCl were added to this catalyst system, good regio- and stereoselectivity were observed. All these results above indicated the crucial role of the Cl ligand in determining the stereochemical outcome of the reaction instead of PPh₃ (Fig. 5D).

**Fig. 4 | Synthetic Utilities. A** gram-scale synthesis. **B** synthetic applications. Reaction conditions: **a** 6 (0.1 mmol, 1.0 equiv.), NIS (0.2 mmol, 2.0 equiv.), 2,6-lutidine (0.7 equiv.), HFIP (0.2 M), 0 °C; **b** 73 (0.1 mmol, 1.0 equiv.), PhB(OH)$_2$ (0.3 mmol, 3.0 equiv.), Pd(PPh$_3$)$_2$Cl$_2$ (10 mol%), SPhos (15 mol%), K$_2$CO$_3$ (0.15 mmol, 1.5 equiv.), Ag$_2$O (0.25 mmol, 2.5 equiv.), 4 Å MS, THF (0.5 M), 60 °C; **c** 73 (0.1 mmol, 1.0 equiv.), phenylacetylene (0.15 mmol, 1.5 equiv.), Pd(PPh$_3$)$_2$Cl$_2$ (5 mol%), CuI (10 mol%), Et$_3$N (0.3 mmol, 3.0 equiv.), DMF, (0.5 M), rt; **d** 73 (0.1 mmol, 1.0 equiv.), ethyl acrylate (0.2 mmol, 2.0 equiv.), Pd(PPh$_3$)$_4$ (5 mol%), Et$_3$N (2.0 equiv.), MeCN (0.5 M), 70 °C; **e** 6 or 3 or 49 (0.1 mmol, 1.0 equiv.), pinnacol (0.5 mmol, 5.0 equiv.), NaHCO$_3$ (0.5 mmol, 5.0 equiv.), CH$_3$OH, 50 °C, then PhCHO (0.11 mmol, 1.1 equiv.), Et$_2$O (0.2 M), rt.; **f** 49 (0.1 mmol, 1.0 equiv.), pinnacol (0.5 mmol, 5.0 equiv.), NaHCO$_3$ (0.5 mmol, 5.0 equiv.), CH$_3$OH, 50 °C, then NaBO$_3$·4H$_2$O (0.3 mmol, 3.0 equiv.), THF/H$_2$O (1:1, 0.2 M), rt; **g** 77 (0.1 mmol, 1.0 equiv.), methyl 4-iodobenzoate (0.15 mmol, 1.5 equiv.), Pd$_2$dba$_3$ (5 mol%), TBAF (2.0 equiv.), THF (0.2 M), 60 °C.

Deuteration experiments with DSiEt$_3$ (99% D) yielded **3-d** and **46-d** smoothly with 99% D-incorporation at the γ-position (Fig. 5E). The kinetic isotope effect measured via intermolecular competition between HSiEt$_3$ and DSiEt$_3$ gave an inverse KIE ($k_H/k_D = 0.67$ for **3** & **3-d**, $k_H/k_D = 0.47$ for **46** & **46-d**) with both Ru catalytic (Fig. 5F). Similar to this result, a parallel kinetic isotope experiment with CpRu(PPh$_3$)$_2$Cl as the catalyst also gave an inverse KIE of 0.32 (Fig. 5G). We proposed that during the concerted oxidative-addition and hydride-insertion, the weaker Si–H/D bond are partially weakened and C–H/D bond are partially formed. Due to the higher bond energy of C–D bond than C–H bond, the driving force of forming C–D bond should be stronger[73–75].

## Computational mechanistic studies

To gain a clearer understanding of the mechanisms underlying the regioselectivity and stereoselectivity in this reaction, DFT calculations were conducted (Fig. 6, Supplementary Data 1 & Supplementary Data 2). In the [Cp*Ru(MeCN)$_3$]PF$_6$-catalyzed system (Fig. 6A), the propargyl MIDA boronate and triethoxysilane reagent initially exchange with the MeCN ligands in the [Cp*Ru(MeCN)$_3$]PF$_6$ catalyst, forming the intermediates **INT-1β** and **INT-1γ**. For β-selectivity, a concerted hydrometallation occurs through transition state **TS-1β** ($\Delta G^{\neq} = 12.3$ kcal/mol), leading to the formation of the ruthenium three-membered ring intermediate **INT-2β-anti** ($\Delta G = 2.5$ kcal/mol). In comparison, achieving γ-selectivity requires a higher activation barrier (via **TS-1γ**, $\Delta G^{\neq} = 14.1$ kcal/mol), and the corresponding intermediate **INT-2γ-anti** is also higher in energy ($\Delta G = 10.7$ kcal/mol). This indicates that the concerted hydrometallation process is the regioselectivity-determining step, with **TS-1γ** and **TS-1β** differing in energy by $\Delta\Delta G^{\neq} = 1.8$ kcal/mol.

NBO orbital interaction analysis reveals that **INT-2β-anti** exhibits a significant orbital interaction between σ(C–B) and π*(C = Ru), with a second-order perturbation energy ($E^{(2)}$) of 7.6 kcal/mol. This suggests that the hyperconjugative effect of the C–B bond significantly stabilizes the C = Ru bond. In contrast, no such orbital interaction is present in **INT-2γ-anti**. Therefore, the hyperconjugative interaction between σ(C–B) and π*(C = Ru) likely plays a crucial role in driving the β-regioselectivity during the concerted hydrometallation process.

Next, **INT-2β-anti** undergoes reductive elimination (via **TS-2β-anti**, $\Delta G^{\neq} = 9.9$ kcal/mol) to yield the *anti*-product **β-PC-anti**. However, to obtain the *syn*-product from **INT-2β-anti**, an additional C$_\beta$–C$_\gamma$ bond rotation is required. Nevertheless, the scanning results reveal that the C$_\beta$–C$_\gamma$ bond directly rotation from **INT-2β-anti** is more likely to lead to retro-hydrosilylation. Since the MeCN can facilely dissociate from the Ru center in **INT-2β-anti** and re-coordinate to the metal center to form **INT-2β'-anti** (see Supplementary Information for details), the **INT-2β'-syn** can be formed by the C$_\beta$–C$_\gamma$ bond rotation from **INT-2β'-anti** (via **TSβ-rot**, $\Delta G^{\neq} = 13.7$ kcal/mol). Although **INT-2β'-syn** is thermodynamically more stable than **INT-2β'-anti** due to steric effects, the energetically demanding **TSβ-rot** prevents the facile conversion of **INT-2β'-anti** to **INT-2β'-syn**. Once **INT-2β'-syn** is formed, it can undergo reductive elimination (via **TS-2β'-syn**, $\Delta G^{\neq} = 7.8$ kcal/mol) to produce the *syn*-product **β-PC-syn**. The energy difference between **TSβ-rot** and **TS-2β-anti** is 3.8 kcal/mol, which aligns with the experimentally observed high stereoselectivity favoring the *anti*-configuration over the *syn*-configuration for the [Cp*Ru(MeCN)$_3$]PF$_6$-catalyzed reaction.

In the CpRu(PPh$_3$)$_2$Cl-catalyzed system (Fig. 6B), initial ligand exchange leads to the formation of intermediates **$^{Cp}$INT-1β** and

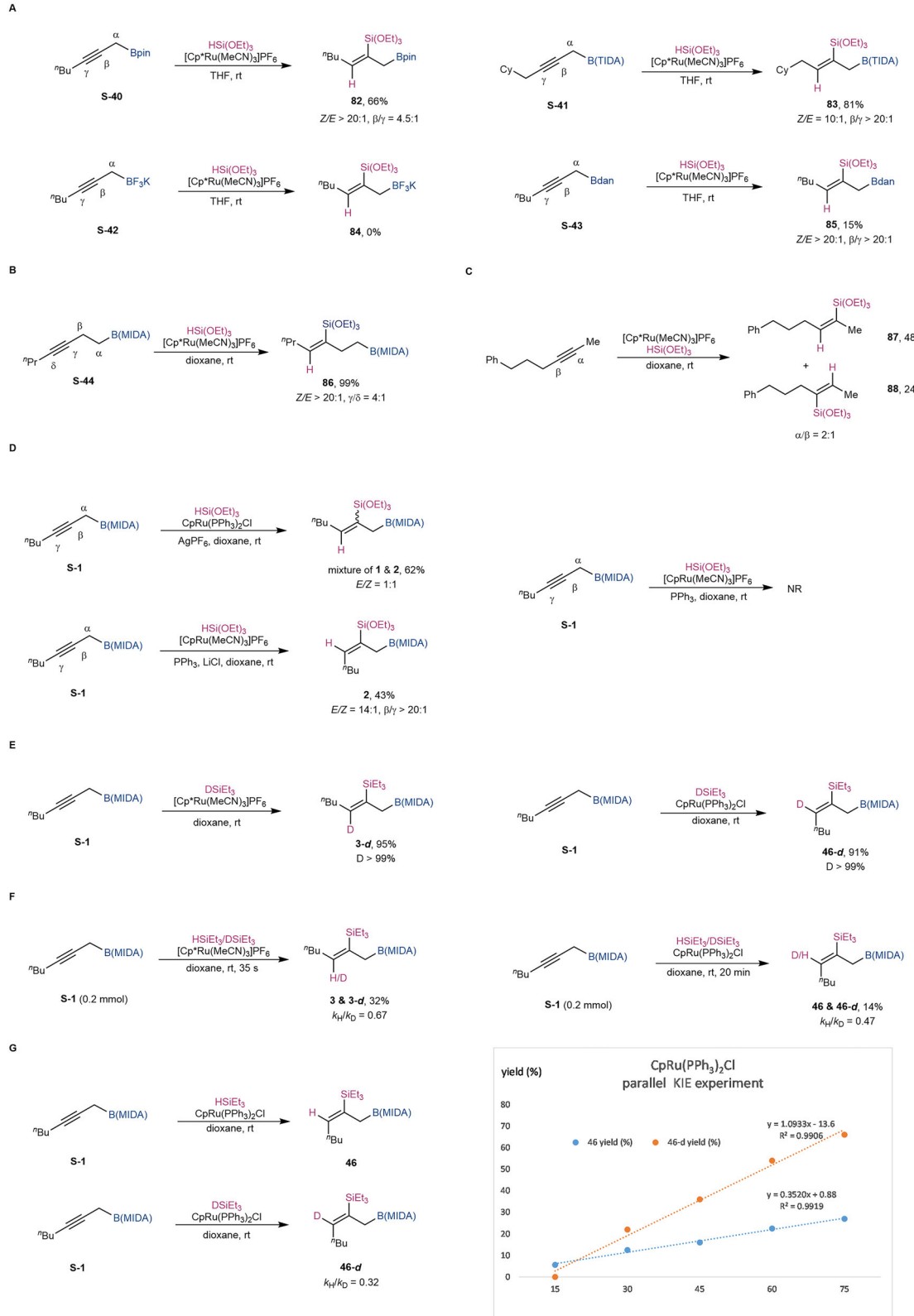

**Fig. 5 | Experimental Mechanistic Studies. A** selectivity of other [B] groups. **B** selectivity of homopropargylic B(MIDA). **C** selectivity of hex-4-yn-1-ylbenzene. **D** selectivity of hex-4-yn-1-ylbenzene. **E** deuteration experiments. **F** intermolecular KIE experiment. **G** parallel KIE experiment.

**CpINT-1γ**. For β-selectivity, a concerted hydrometallation occurs via **CpTS-1β** (ΔG‡ = 22.9 kcal/mol), forming the three-membered ring intermediate **CpINT-2β-syn**. Similar to the [Cp*Ru(MeCN)₃]PF₆-catalyzed system, the γ-site selective concerted hydrometallation **CpTS-1γ** requires a higher energy barrier (ΔG‡ = 27.3 kcal/mol) to form the

intermediate **CpINT-2γ-anti**, which is also higher in energy (ΔG = 19.9 kcal/mol). NBO orbital interaction analysis indicates that **CpINT-2β-syn** exhibits a significant orbital interaction between σ(C−B) and π*(C = Ru) (E⁽²⁾ = 10.1 kcal/mol), whereas **CpINT-2γ-anti** does not show such interaction. Thus, the β-regioselectivity is attributed to

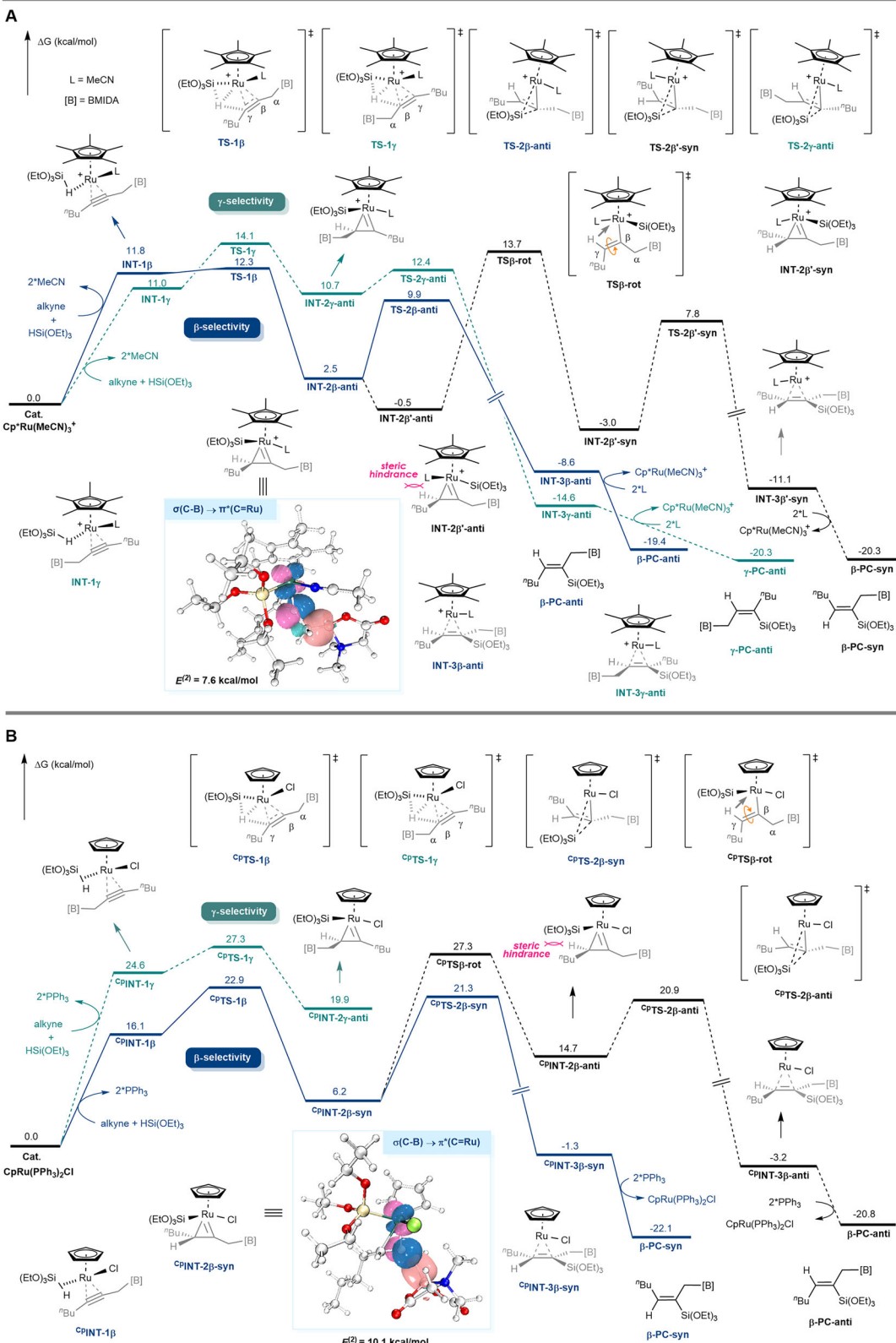

**Fig. 6 | Computational Studies. A** the [Cp*Ru(MeCN)$_3$]PF$_6$-catalyzed system. **B** the CpRu(PPh$_3$)$_2$Cl-catalyzed system.

the hyperconjugative effect between σ(C–B) and π*(C=Ru). Following this, **$^{Cp}$INT-2β-syn** undergoes reductive elimination via **$^{Cp}$TS-2β-syn** (ΔG$^{\neq}$ = 21.3 kcal/mol), yielding the **syn-product β-PC-syn**. The high activation barrier of **$^{Cp}$TSβ-rot** (ΔG$^{\neq}$ = 27.3 kcal/mol) prevents the rotation of the C$_β$–C$_γ$ bond, despite the subsequent reductive elimination via **$^{Cp}$TS-2β-anti** being relatively low in

energy (ΔG$^{\neq}$ = 20.9 kcal/mol). The significant energy difference of 6.0 kcal/mol between **$^{Cp}$TSβ-rot** and **$^{Cp}$TS-2β-syn** aligns with the experimentally observed high stereoselectivity between the *anti* and *syn* configurations for CpRu(PPh$_3$)$_2$Cl. It was found that Cl$^-$ plays a crucial role in stereoselectivity by imposing steric hindrance (see the Supplementary Information for further details).

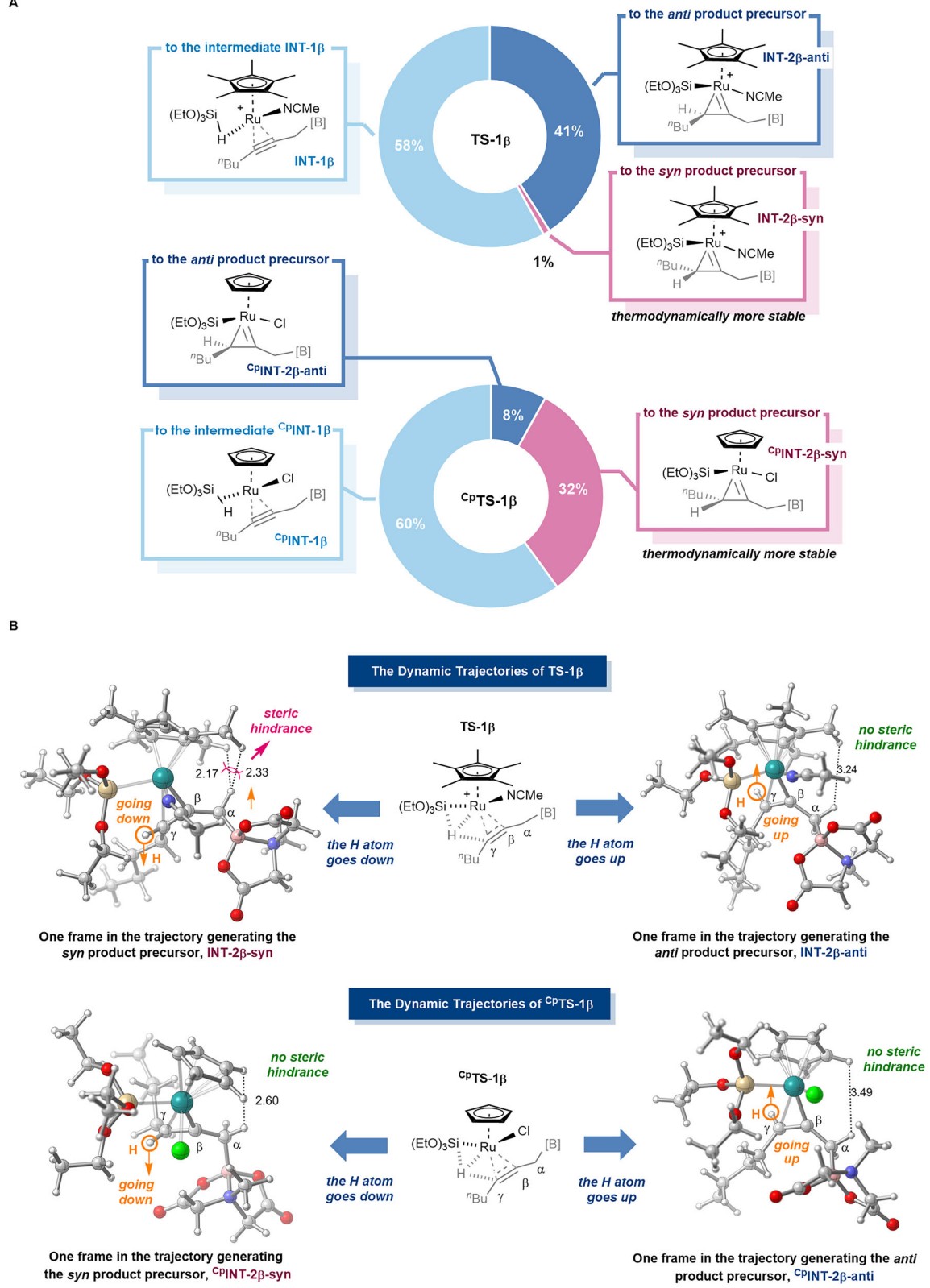

**Fig. 7 | The Results of BOMD Calculations. A** Sum of the BOMD Results. **B** Conformational Analysis of the Dynamic Trajectories.

The key distinction between the [Cp*Ru(MeCN)₃]PF₆-catalyzed system and the CpRu(PPh₃)₂Cl-catalyzed system lies in their respective concerted hydrometallation processes. Specifically, in the [Cp*Ru(MeCN)₃]PF₆-catalyzed system, the transition state **TS-1β** results in the generation of **INT-2β-anti**, the *anti*-product precursor. Conversely, in the CpRu(PPh₃)₂Cl-catalyzed system, the transition state **ᶜᵖTS-1β** leads

to the formation of **ᶜᵖINT-2β-syn**, the *syn*-product precursor. To investigate the underlying reasons behind the *anti/syn* selectivity in these different catalytic systems, we performed Born-Oppenheimer molecular dynamics (BOMD) calculations[76,77] on the transition states **TS-1β** and **ᶜᵖTS-1β** (Fig. 7A). For the [Cp*Ru(MeCN)₃]PF₆-catalyzed system, among 100 dynamic trajectories, 58 led to **INT-1β** preceding

**TS-1β**, 41 resulted in the *anti*-product precursor **INT-2β-anti**, and only one yielded the *syn*-product precursor. In contrast, for the CpRu(PPh₃)₂Cl-catalyzed system, 32 out of 100 trajectories produced the *syn*-product precursor **ᶜᵖINT-2β-syn**, while only 8 led to the *anti*-product precursor. These BOMD results align with the reaction potential energy surfaces described above, indicating that the *anti/syn* selectivity in these catalytic processes is kinetically controlled.

The detailed conformational analysis of dynamic trajectories based on **TS-1β** and **ᶜᵖTS-1β** is depicted in Fig. 7B. For the [Cp*Ru(MeCN)₃]PF₆ catalyst, when the H atom moves downward, it induces significant steric hindrance between the CH₂–B(MIDA) group and the bulky Cp ring (the distance between the nearest two H atoms is 2.17 Å, which is shorter than the sum of the van der Waals radii of two H atoms, 2.40 Å). This hindrance arises because, prior to the formation of the three-membered ring intermediate, both the H atom and the CH₂-B(MIDA) group are connected to the same plane of the double bond. Consequently, as the H atom moves downward, the CH₂–B(MIDA) group shifts upward, drawing closer to the Cp* ring. Conversely, in dynamic trajectories where the H atom moves upward, it does not cause associated steric hindrance, making this pathway more favorable and leading to the formation of the *anti*-product precursor **INT-2β-anti**. In contrast, with the CpRu(PPh₃)₂Cl catalyst, minimal or no steric hindrance from the Cp ligand is observed, regardless of whether the H atom moves upward or downward to form the three-membered ring. Consequently, the dynamic trajectory leading to the formation of the thermodynamically more stable *syn*-product precursor **ᶜᵖINT-2β-syn** is favored.

To sum up, the DFT calculation results indicate that concerted hydrometallation is the determining step for regioselectivity and stereoselectivity. The hyperconjugative effect of the C−B bond[50,52] significantly stabilizes the C = Ru bond at the β position of the boron atom, a crucial factor influencing β-regioselectivity. Meanwhile, the differing steric effects of Cp* and Cp in different catalytic systems contribute to the stereoselectivity of this reaction.

## Discussion

In summary, we have demonstrated the significant impact of the β-boron effect in achieving regioselective Ru-catalyzed hydrosilylation of propargylic B(MIDA)s. This approach leverages the interaction between the σ(C−B) orbital and the electrophilic metallacyclopropene intermediate, akin to a Fisher carbene, to achieve high regioselectivity. Additionally, subtle variations in the Ru catalyst have led to a switch in stereoselectivity without affecting regioselectivity, showcasing a rare instance of stereo-divergence in metal-catalyzed hydrosilylation. Our work introduces a versatile synthesis route for regio- and stereo-defined building blocks that incorporate boryl, silyl, and alkene functionalities. This study paves the way for future exploration of β-boryl effect-guided strategies in metal-catalyzed transformations, offering avenues for the precise and efficient synthesis of complex organic molecules.

## Methods

### General procedure A for the synthesis of products in *Z* isomers

In the glove box, to an oven-dried 15 mL vial were added the propargylic MIDA boronate (0.10 mmol), the silane (0.20 mmol), [Cp*Ru(MeCN)₃]PF₆ (2.5 mg, 5.0 μmol), and THF or 1,4-dioxane (1.0 mL). The vial was capped and removed from the glove box. The reaction mixture was stirred at room temperature for 30 min, and then concentrated under reduced pressure. The residue was purified by silica gel flash column chromatography (ethyl acetate–petroleum as the eluent) to give the desired product.

### General procedure B for the synthesis of products in *E* isomers

In a glove box, to an oven-dried 15 mL vial were added the propargylic MIDA boronate (0.10 mmol), the silane (0.20 mmol), CpRu(PPh₃)₂Cl (3.6 mg, 5.0 μmol), and 1,4-dioxane (1.0 mL). The vial was capped and

removed from the glove box. The reaction mixture was stirred at room temperature for 3 h, and then concentrated under reduced pressure. The residue was purified by silica gel flash column chromatography (ethyl acetate–petroleum as the eluent) to give the desired product.

## Data availability

The authors declare that the main data supporting the findings of this study, including experimental procedures, characterization of materials and products, general methods, and NMR spectra, are available within the article and its Supplementary Information. All data are available from the corresponding author upon request.

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

## Acknowledgements

This work was supported by Noncommunicable Chronic Diseases-National Science and Technology Major Project (2023ZD0507600 to H. W.), the National Natural Science Foundation of China (22371311 to H.W., 22403067 to T.-Y.S. and 22171293 to Q.L.), Guangdong Provincial Key Laboratory of Construction Foundation (2023B1212060022 to Q.L.), Guangzhou Municipal Science and Technology Project (2025A04J7101 to H.W.), Shenzhen Bay Laboratory High Performance Computing and Informatics Facility, Open Research Fund of State Key Laboratory of Coordination Chemistry, School of Chemistry and Chemical Engineer-ing, Nanjing University, and the GBRCE for Functional Molecular Engineering.

## Author contributions

J.Q. and S.L. contributed equally. J. Q. conducted reaction optimization and contributed to the substrate scope. J.Q. and Q.L. contributed to the analysis of the data. S. L. and T.-Y.S. performed the DFT calculations. J.Q., S.L., T.-Y.S., and H.W. wrote the paper. Z.-H.C., J.H., and W.Z. contributed to the synthesis of substrates. All authors contributed to discussions.

## Competing interests

The authors declare no competing interests.
