## [Transparent Peer Review file · Nature Communications]

Harnessing the β -Boron Effect for Regioselective Ru-Catalyzed Hydrosilylation of Internal Alkynes

Corresponding Author: Professor Honggen Wang

Version 0:

Reviewer comments:

Reviewer #1

(Remarks to the Author)

In this manuscript, Wang, Sun, and co-workers reported a new strategy for beta-boron effect-guided regioselective Ru-catalyzed hydrosilylation of propargylic B(MIDA)s. The high regioselective of the reaction was attributed to the profound directing effect of the B(MIDA) moiety through the hyperconjugative effect of the C-B bond. Besides, subtle variations in the Ru catalyst can lead to a remarkable switch in stereoselectivity, which was related to the ligand steric effects of Cp* and Cp. Finally, detailed DFT calculation and BOMD calculation were carried out to reveal the regioselectivity and stereoselectivity of the reaction.

Above all, this paper appears to have been convincingly carried out and with good novelty, and in my opinion, this paper is a nice piece of work that merits publication in Nat. Commun.

However, some issues should be addressed by the authors before the publication of this work.

1. There are many detail errors in the article, please check carefully. For example, the second paragraph on page 2 (Line 49-61) was the same as the second paragraph on page 2 (Line 33-45), please check.
2. On page 4, line 97, the first [CpRu(MeCN)₃]PF₆ should be corrected to [Cp Ru(MeCN)₃]PF₆.
3. In Scheme 3, the Z isomer should be corrected to E isomer.
4. On page 8, Line 194, it mentioned that the crucial role of the Cl ligand in determining the stereochemical outcome of the reaction. However, there is no mention of the role of Cl atom in the computational studies section.
5. In the article, authors performed Born-Oppenheimer molecular dynamics (BOMD) calculations, some relevant references should be cited to confirm the reliability of this method.
6. In Supporting information, the range of the ¹H NMR spectra of compounds should be given to 10 ppm, please redraw. Besides, the ¹¹B NMR spectra of all compounds were missing.
7. In Supporting information, on page 10, the ¹H NMR spectra of S-6: 1.89 (p, J= 6.7 Hz, 2H), what does p stand for? The same spectradescription appears in compounds S-34 (Page 17), compound 14 (Page 25), 26 (Page 32), compound 30 (Page 34), compound 48 (Page 44), compound 52 (Page 46), compound 54 (Page 47), compound 65 (Page 53) and so on.
8. In Supporting information, the noise signal of the NOE spectra was too high, please reprocess. For example: compound 3 (Page 149), 15 (Page 162), 17 (Page 164), 22 (Page 172), 23 (Page 173), 26 (Page 177), 37 (Page 189).

Reviewer #2

(Remarks to the Author)

Comments:

The authors present in this paper the regioselective Ru-catalyzed hydrosilylation of propargylic B(MIDA) compounds to synthesize alkenylsilanes. The switchable stereoselectivity resulted from different catalysts [Cp*₂Ru(MeCN)₃]PF₆ and CpRu(PPh₃)₂Cl is interesting. However, the explanation of stereoselectivity and regioselectivity given by the authors contains many oversights. Specifically, their assertion that the hyperconjugative effect of the σ (C-B) bond is crucial for achieving high regioselectivity may lack rigor. Consequently, the authors must address the following questions to clarify their findings. The reviewer will determine the suitability of the paper for publication in Nature Communications based on the authors' responses.

1. Very surprised that the third paragraph of the introduction repeats the second, a very serious low-level error.

2. The "β-Boron Effect" claimed by the authors appears to be a concept. However, it is not rigorous for the authors to propose the "β-Boron Effect" without trying various other B-group substituents.
3. (The first paragraph of the "Results and Discussion" section) The statement "but MeCN proved unsuitable as a solvent likely due to its strong coordinating nature (entry 3)" raises the question of what is the basis for such an inference and what is the specific manifestations of its strong coordinating ability.
4. (The first paragraph of the "Results and Discussion" section) "An intriguing observation arose when we substituted [CpRu(MeCN)₃]PF₆ with its analogue [CpRu(MeCN)₃]PF₆, featuring a smaller Cp ligand". Please check the writing here.
5. The yields of terminal alkynes 41 and 42, as indicated in Scheme 2, are unexpectedly low. What accounts for this phenomenon? Furthermore, disregarding the issue of stereoselectivity, does this hydrosilylation method have a restricted applicability solely to internal alkynes?
6. For compounds 59, 61, and 62, the authors state, "These results underscore the significant directing influence of the B(MIDA) moiety." To justify such a conclusion, the authors should investigate the role of the directing group in 59, 61, and 62 through DFT calculations.
7. In the "Experimental Mechanistic Studies" section, the authors present the experimental results of using sp²-B propargylic Bpin S-37 in the reaction, which showed a significantly reduced β/γ ratio of 4:1. However, no relevant explanation was provided for this observation. What does this result imply? It is recommended to conduct DFT calculations to compare the differences between Bpin and B(MIDA).
8. In the "Experimental Mechanistic Studies" section, it is suggested to investigate asymmetric internal alkyne substrates with B(MIDA) removed as a comparative experiment.
9. In the first paragraph of the "Computational Mechanistic Studies" section, the ΔΔG[‡] value of 1.8 kcal/mol is too small to account for the excellent β-selectivity observed with a β/γ ratio of greater than 20:1.
10. In Scheme 6A, considering that the γ-product is a genuinely existing byproduct, the authors should provide a complete mechanism for the formation of the γ-product rather than stopping at INT-2γ-anti.
11. (The second paragraph of the "Computational Mechanistic Studies" section) TS-1γ and TS-1β are crucial in determining the selectivity. The authors should focus on analyzing these two transition states. Please provide the reasons for the energy difference and explain whether it is related to the interaction between σ(C-B) and π*(C=Ru).
12. In Scheme 6A, INT-2β-anti undergoes isomerization to INT-2β'-anti, followed by rotation of the Cβ-Cγ bond. The isomerization from INT-2β-anti to INT-2β'-anti involves an exchange of positions between the L ligand and the -Si(OEt)₃ group, which seems improbable. The authors are requested to provide detailed computational results of the isomerization process.
13. In Scheme 6A, the authors are requested to present the mechanistic process of direct Cβ-Cγ bond rotation from INT-2β-anti. This represents the most straightforward mechanism, so why was it not considered by the authors? The authors should explain the necessity of INT-2β'-anti as an intermediate in the mechanism.
14. What does the double wavy line between TS-2β-anti and INT-3β-anti represent? If it means that some steps have been omitted between the two, please give details of the steps in the supporting material.
15. Similar to comment 11, for Scheme 6B, the authors claim that the β-regioselectivity is attributed to the hyperconjugative effect between σ(C-B) and π*(C=Ru). To assess the rationality of this inference, it is imperative to examine CpTS-1γ and CpTS-1β, rather than solely focusing on the non-selectivity-determining intermediates CpINT-2β-syn and CpINT-2γ-anti.
16. The authors should attempt to rotate the H-Si(OEt)₃ group in CpINT-1β to form an intermediate where the hydrogen is oriented inwards. Furthermore, they should investigate the possibility of obtaining CpINT-2β-anti from this intermediate.
17. Based on the experimental results shown in entry 5 of Table 1 (where the MeCN ligand is replaced with PPh₃ in the presence of Cp, leading to an increase in E-selectivity) and entry 6 (where the ligand is changed to PPh₃ in the presence of Cp*, resulting in an improvement in Z-selectivity), it is evident that, apart from Cp and Cp*, the ligands MeCN and PPh₃ also influence stereoselectivity. The authors are requested to analyze and explain the impact of the differences between MeCN and PPh₃ on stereoselectivity, an important aspect that is not addressed at all in the paper.

Reviewer #3

(Remarks to the Author)

The manuscript by Wang, Sun, and coworkers describes a Ru-catalyzed regio- and stereoselective hydrosilylation of propargylic B(MIDA). Transition metal-catalyzed hydrosilylation of alkynes offers a direct and atom-economical approach to synthesizing alkenylsilanes, a family of useful compounds. However, current state-of-the-art methods primarily focus on the

hydrosilylation of terminal alkynes. The use of unsymmetrical internal alkynes has faced challenges in achieving precise regio- and stereocontrol. Here, the authors achieve regioselective hydrosilylation of internal alkynes under Ru catalysis by harnessing the β -boron effect. Stereodivergent control is also realized by subtle variations in the Ru catalyst, which enables the synthesis of diverse regio- and stereo-defined building blocks that amalgamate the synthetic potential of boryl, silyl, and alkene functionalities. Control experiments and DFT studies were conducted to shed light on the detailed mechanism. Overall, I support the acceptance of the manuscript if the following points are sufficiently addressed.

1. The content in the third paragraph of the Introduction repeats information already presented in the second paragraph.
2. For the syn-hydrosilylation reaction, $\text{CpRu}(\text{PPh}_3)_2\text{Cl}$ has been identified as the optimal catalyst. Does the presence of PPh_3 have any significant impact on the reaction? It would be beneficial to conduct control experiments to know more about this point.
3. Regarding the substrate scope, could aromatic alkynes and enynes be applicable to this reaction? Exploring these substrates would help broaden the scope and applicability of the reaction.
4. In Scheme 3, the labeling of the products in the model reaction, 'Z isomer' should be 'E isomer'. Please check.
5. It would be valuable to examine a gram-scale experiment to evaluate the robustness and practicality of the reaction on a larger scale.
6. In the DFT calculation section, more transition states should be considered. Specifically, the transition state involving planar σ -vinyl intermediates should be included, particularly from $\text{TS-1}\beta$ to $\text{INT-2}\beta$ -anti. In this case, two different ruthenacyclopentene intermediates can be formed through clockwise or counterclockwise rotation of the σ -vinyl ruthenium. Similarly, calculations for the reaction involving syn-selectivity should also be conducted.
7. To further validate the computational results, the authors are encouraged to perform additional mechanistic experiments, such as deuteration experiments, kinetic isotope effect (KIE) studies, and kinetic studies.

For SI:

1. The mass of the products should be included in the SI, rather than just providing the yield.
2. Page 43 in the SI, product 47, 'calculated for ...Si...' should be 'calculated for ...Si₃(subscript)...'.
3. Page 43 in the SI, product 48, there is no methyl group information in the ¹H NMR data.
4. Page 174 in the SI, for product 24, the chemical shifts in the copy of ¹H NMR spectrum should be masked, and the chemical shifts in the ¹³C NMR spectrum are inconsistent with the corresponding data on page 31.

Version 1:

Reviewer comments:

Reviewer #1

(Remarks to the Author)

The authors have addressed the issues and questions raised by the reviewers in the first round, and it is could be accepted at this stage.

Reviewer #2

(Remarks to the Author)

The authors have adequately addressed the concerns I raised, and I have no further questions.

Reviewer #3

(Remarks to the Author)

The authors have made suitable revisions and I support the acceptance of this manuscript.

point-by-point response

REVIEWER COMMENTS

Reviewer #1 (Remarks to the Author):

In this manuscript, Wang, Sun, and co-workers reported a new strategy for beta-boron effect-guided regioselective Ru-catalyzed hydrosilylation of propargylic B(MIDA)s. The high regioselective of the reaction was attributed to the profound directing effect of the B(MIDA) moiety through the hyperconjugative effect of the C-B bond. Besides, subtle variations in the Ru catalyst can lead to a remarkable switch in stereoselectivity, which was related to the ligand steric effects of Cp* and Cp. Finally, detailed DFT calculation and BOMD calculation were carried out to reveal the regioselectivity and stereoselectivity of the reaction.

Above all, this paper appears to have been convincingly carried out and with good novelty, and in my opinion, this paper is a nice piece of work that merits publication in Nat. Commun. However, some issues should be addressed by the authors before the publication of this work.

1. There are many detail errors in the article, please check carefully. For example, the second paragraph on page 2 (Line 49-61) was the same as the second paragraph on page 2 (Line 33-45), please check.

Response: Thanks. We have deleted the repeating paragraph.

2. On page 4, line 97, the first [CpRu(MeCN)₃]PF₆ should be corrected to [Cp*Ru(MeCN)₃]PF₆.

Response: Thanks, revised.

3. In Scheme 3, the Z isomer should be corrected to E isomer.

Response: Thanks, revised.

4. On page 8, Line 194, it mentioned that the crucial role of the Cl ligand in determining the stereochemical outcome of the reaction. However, there is no mention of the role of Cl atom in the computational studies section.

Response:

Thank you for highlighting this important scientific issue. Following your suggestion, we conducted additional DFT calculations. As illustrated in the figure below, the intermediates **^{Cp}INT-2β-syn** and **^{Cp}INT-2β-anti** exhibit excellent stereochemical selectivity, with an energy difference of 8.5 kcal/mol. Upon removal of the Cl ligand from **^{Cp}INT-2β-syn** and **^{Cp}INT-2β-anti**, the resulting isomers, **^{Cp}INT-2β-syn-no-Cl** and **^{Cp}INT-2β-anti-no-Cl**, exhibit nearly identical energies, differing by only 0.4 kcal/mol, indicating poor selectivity.

The introduction of the Cl ligand increases structural crowding, particularly in **^{Cp}INT-2β-anti**. In the absence of the Cl ligand, the distance between the butyl chain carbon and

the Si atom in $\text{Cp}^{\text{PINT-2}\beta\text{-anti-no-Cl}}$ is 3.94 Å. However, with the Cl ligand present, this distance decreases to 3.48 Å in $\text{Cp}^{\text{PINT-2}\beta\text{-anti}}$, which is less than the sum of the van der Waals radii of C and Si (3.80 Å). This results in significant steric hindrance, leading to a higher energy for $\text{Cp}^{\text{PINT-2}\beta\text{-anti}}$.

These results were included in SI (page 78) and commented in the main text.

5. In the article, authors performed Born-Oppenheimer molecular dynamics (BOMD) calculations, some relevant references should be cited to confirm the reliability of this method.

Response:

Thank you for your constructive suggestion. Two relevant references have been cited in the main text (J. Am. Chem. Soc. 139, 1726-1729 (2017); Chem 4, 1952-1966 (2018)). Born-Oppenheimer Molecular Dynamics (BOMD) is frequently employed in scenarios where two distinct products can be generated from the same transition state. This approach allows for the exploration of reaction pathways that lead to different outcomes, providing valuable insights into the reaction mechanisms and product distributions.

6. In Supporting information, the range of the ^1H NMR spectra of compounds should be given to 10 ppm, please redraw. Besides, the ^{11}B NMR spectra of all compounds were missing.

Response:

Thank you for your advice. The range of the ^1H NMR spectra has been given to 10 ppm.

The ^{11}B NMR chemical shift for B(MIDA)s is insensitive to the structure. We thus provide only the ^{11}B NMR for some representative cases (**S-42**, **3**, **6**, **13**, **22**, **23**, **54**, **70**, **73**) in SI. And indeed, the peaks of ^{11}B NMR (B(MIDA)s) range in 11.8-12.8 ppm.

7. In Supporting information, on page 10, the ^1H NMR spectra of S-6: 1.89 (p, J= 6.7 Hz,

2H), what does p stand for? The same spectra description appears in compounds **S-34** (Page 17), compound **14** (Page 25), **26** (Page 32), compound **30** (Page 34), compound **48** (Page 44), compound **52** (Page 46), compound **54** (Page 47), compound **65** (Page 53) and so on.

Response: You are right. These signals should be tt or m. The spectra descriptions are corrected.

8. In Supporting information, the noise signal of the NOE spectra was too high, please reprocess. For example: compound 3 (Page 149), 15 (Page 162), 17 (Page 164), 22 (Page 172), 23 (Page 173), 26 (Page 177), 37 (Page 189).

Response:

Thank you for your advice. We have reprocessed the recordings for compounds 3 and 23. The new NOE spectra have been provided, but they do not seem to show significant improvement.

Reviewer #2 (Remarks to the Author):

Comments:

The authors present in this paper the regioselective Ru-catalyzed hydrosilylation of propargylic B(MIDA) compounds to synthesize alkenylsilanes. The switchable stereoselectivity resulted from different catalysts [Cp*Ru(MeCN)₃]PF₆ and CpRu(PPh₃)₂Cl is interesting. However, the explanation of stereoselectivity and regioselectivity given by the authors contains many oversights. Specifically, their assertion that the hyperconjugative effect of the $\sigma(\text{C-B})$ bond is crucial for achieving high regioselectivity may lack rigor. Consequently, the authors must address the following questions to clarify their findings. The reviewer will determine the suitability of the paper for publication in Nature Communications based on the authors' responses.

1. Very surprised that the third paragraph of the introduction repeats the second, a very serious low-level error.

Response: Thanks. We have deleted the repeating paragraph.

2. The “ β -Boron Effect” claimed by the authors appears to be a concept. However, it is not rigorous for the authors to propose the “ β -Boron Effect” without trying various other B-group substituents.

Response:

Thank you for your constructive comments. “ β -Boron effect” has been confirmed experimentally and theoretically in our previous studies (*J. Am. Chem. Soc.* **2022**, *144*, 14380; *J. Am. Chem. Soc.* **2023**, *145*, 7548; *Adv. Sci.* **2023**, *10*, 2304282.)

Based on your suggestion, we explored several other commonly encountered boron species in our reaction. Notably, the use of sp²-B propargylic Bpin (**S-40**) predominantly afforded the β -silylated product with a β/γ ratio of 4:1. As anticipated, B(TIDA) exhibited excellent β -selectivity. In contrast, BF₃K showed no reactivity, even when the reaction was

performed in various solvents to rule out solubility issues. Bdan also demonstrated excellent β -selectivity, though with a relatively low yield.

Taken together, these results suggest that the β -boron effect may be present in multiple boron species, with B(MIDA) delivering the most favorable outcomes.

These results and comments were added to the manuscript.

a) selectivity of other [B] groups

3. (The first paragraph of the "Results and Discussion" section) The statement "but MeCN proved unsuitable as a solvent likely due to its strong coordinating nature (entry 3)" raises the question of what is the basis for such an inference and what is the specific manifestations of its strong coordinating ability.

Response: MeCN is a coordinating solvent, as also observed in the work of Sun, Wu, and Lung (Ref. 67, Entry 12 in Table 1). In their study, MeCN effectively suppressed reactivity.

4. (The first paragraph of the "Results and Discussion" section) "An intriguing observation arose when we substituted [CpRu(MeCN)₃]PF₆ with its analogue [CpRu(MeCN)₃]PF₆, featuring a smaller Cp ligand". Please check the writing here.

Response: Thanks, a star was added to the first Cp as Cp*.

5. The yields of terminal alkynes 41 and 42, as indicated in Scheme 2, are unexpectedly low. What accounts for this phenomenon? Furthermore, disregarding the issue of stereoselectivity, does this hydrosilylation method have a restricted applicability solely to internal alkynes?

Response: The low yields of terminal alkynes 41 and 42 are attributed to the incomplete conversion of the starting materials, with approximately 50% remaining unreacted. This may result from catalyst deactivation caused by the presence of the terminal alkynyl C–H bond. A comment was added to the manuscript.

6. For compounds 59, 61, and 62, the authors state, "These results underscore the significant directing influence of the B(MIDA) moiety." To justify such a conclusion, the authors should investigate the role of the directing group in 59, 61, and 62 through DFT calculations.

Response:

Thank you for your insightful comment. We conducted DFT calculations on compounds **59**, **61**, and **62** (as shown in the figure below), which revealed that the transition states influenced by the directing effect of the B(MIDA) moiety (**59-C^pTs-1 β** , **61-C^pTs-1 β** , and **62-C^pTs-1 β**) exhibit energies 2.7 to 4.8 kcal/mol lower than those by hydrogen bonding

(**59-C^pTS-1 γ** , **61-C^pTS-1 γ** , and **62-C^pTS-1 γ**). These results align with the experimentally observed selectivity and further support the presence of the directing effect of the B(MIDA) moiety.

These results were added to the supporting information (page 79) and a comment was added to the reference part in the manuscript.

7. In the "Experimental Mechanistic Studies" section, the authors present the experimental results of using sp^2 -B propargylic Bpin S-37 in the reaction, which showed a significantly reduced β/γ ratio of 4:1. However, no relevant explanation was provided for this observation. What does this result imply? It is recommended to conduct DFT calculations to compare the differences between Bpin and B(MIDA).

Response: Following your suggestion, we conducted additional DFT calculations. As illustrated in the figure below, when replacing B(MIDA) with sp^2 -B Bpin moiety in transition states **TS-1 β** and **TS-1 γ** , the energy difference between **Bpin-TS-1 β** and **Bpin-TS-1 γ** drops from 1.8 kcal/mol to 0.9 kcal/mol, consistent with the experimentally observed reduction in the β/γ ratio to 4:1. Furthermore, compared to the significant energy difference ($\Delta\Delta G = 8.2$ kcal/mol) between **INT-2 β -anti** and **INT-2 γ -anti**, the energies of **Bpin-INT-2 β -anti** and **Bpin-INT-2 γ -anti** are much closer ($\Delta\Delta G = 2.0$ kcal/mol). Specifically, as the figure shown below, the smaller size and lower steric hindrance of Bpin may allow for greater flexibility in the transition state **Bpin-TS-1 γ** (lower steric hindrance), leading to a reduced

energy difference between **Bpin-TS-1 β** and **Bpin-TS-1 γ** . In contrast, the bulkier B(MIDA) group likely imposes stronger steric hindrance (as in **TS-1 γ**), favoring the β -product more selectively.

The NBO orbital interaction analysis was also conducted to clarify the reason for this reduced energy difference. Due to the large energy difference between **Bpin-INT-2 β -anti** and **Bpin-INT-2 γ -anti** (compared to **Bpin-TS-1 β** and **Bpin-TS-1 γ**), it is inferred that their orbital interactions should also differ significantly. Therefore, we chose to perform NBO orbital interaction analysis on the ruthenium three-membered ring intermediates **Bpin-INT-2 β -anti** and **Bpin-INT-2 γ -anti**. For the orbital interactions between $\sigma(\text{C-B})$ and $\pi^*(\text{C=Ru})$, the E^2 energy for **Bpin-INT-2 β -anti** (6.1 kcal/mol) is lower than that for **INT-2 β -anti** (7.6 kcal/mol). Additionally, for the orbital interactions between $\sigma(\text{C-C})$ and $\pi^*(\text{C=Ru})$, while no E^2 value is observed for **INT-2 γ -anti**, an E^2 energy of 3.4 kcal/mol is present in **Bpin-INT-2 γ -anti**. Compared to **INT-2 β -anti** ($E^2 = 7.6$ kcal/mol) and **INT-2 γ -anti** (no E^2 value), the orbital interaction energy difference between **Bpin-INT-2 β -anti** ($E^2 = 6.1$ kcal/mol) and **Bpin-INT-2 γ -anti** ($E^2 = 3.4$ kcal/mol) is much smaller, which also indicates that B(MIDA) and Bpin exhibit different abilities to influence regioselectivity through $\sigma(\text{C-B})$ hyperconjugation.

Overall, based on the calculations results above, the significantly reduced β/γ ratio of 4:1 with Bpin (compared to the higher ratio observed with B(MIDA)) suggests that the steric and electronic properties of the boron substituent play a critical role in determining the regioselectivity of the reaction.

These results were included in the SI (page 79).

8. In the "Experimental Mechanistic Studies" section, it is suggested to investigate asymmetric internal alkyne substrates with B(MIDA) removed as a comparative experiment.

Response:

Thank you for your advice. As shown in **Scheme 5c**, the hydrosilylation of hex-4-yn-1-ylbenzene, an internal alkyne without boron groups, showed a low regioselectivity of 2:1 (**87**, **88**) with $\text{Cp}^*\text{Ru}(\text{MeCN})_3\text{PF}_6$ as the catalyst. It should also be mentioned that, when using $\text{CpRu}(\text{PPh}_3)_2\text{Cl}$ as the catalyst, the hydrosilylation of hex-4-yn-1-ylbenzene showed both low regio- and stereoselectivity, forming four isomers (Z- α , Z- β , E- α , E- β).

These results were included in Scheme 5c.

9. In the first paragraph of the "Computational Mechanistic Studies" section, the $\Delta\Delta G^\ddagger$ value of 1.8 kcal/mol is too small to account for the excellent β -selectivity observed with a β/γ ratio of greater than 20:1.

Response:

Thanks for pointing out this issue. In this study, we applied the Boltzmann distribution equation to calculate the ratio between the two states:

$$N_B/N_A = \exp(-\Delta\Delta G^\ddagger/RT)$$

Where:

$\Delta\Delta G$ is the free energy difference between the two products (in this case, 1.8 kcal/mol)

R is the universal gas constant, which is approximately 1.987 cal/(mol·K)

T is the temperature in Kelvin (room temperature, 298 K)

Thus:

$$N_B/N_A = \exp(-3.03) \approx 0.047$$

This means that the ratio of product B to product A is approximately 0.047 (about 1:21).

According to the calculated ratio above, the value of 1.8 kcal/mol is in a good line with the excellent β -selectivity observed with a 20:1 β/γ ratio.

10. In Scheme 6A, considering that the γ -product is a genuinely existing byproduct, the authors should provide a complete mechanism for the formation of the γ -product rather than stopping at INT-2 γ -anti.

Response:

Thanks for your kind suggestion. The pathway from **INT-2 γ -anti** to the γ -product has been considered. Please see the details in the revised Scheme 6A or the figure below.

A. The $[\text{Cp}^*\text{Ru}(\text{MeCN})_3]\text{PF}_6^-$ -Catalyzed System

11. (The second paragraph of the "Computational Mechanistic Studies" section) TS-1 γ and TS-1 β are crucial in determining the selectivity. The authors should focus on analyzing these two transition states. Please provide the reasons for the energy difference and explain whether it is related to the interaction between $\sigma(\text{C-B})$ and $\pi^*(\text{C=Ru})$.

Response:

Thank you for your thoughtful consideration. According to the Hammond postulate (*J. Am. Chem. Soc.* **1955**, *77*, *2*, 334–338), in endothermic reactions, the transition state structures more closely resemble the products, and the greater the stability of the products, the lower the energy of the transition state.

Hammond postulate

Since the energy difference between the products INT-2 β -anti and INT-2 γ -anti is more significant and follows a similar trend to that of the transition states TS-1 β and TS-1 γ , analyzing the key three-membered ring intermediates INT-2 β -anti and INT-2 γ -anti can

provide valuable insight into why **TS-1 β** is more stable.

Our NBO analysis of the key molecular orbitals in **INT-2 β -anti** and **INT-2 γ -anti** revealed that **INT-2 β -anti** exhibits a significant orbital interaction between $\sigma(\text{C-B})$ and $\pi^*(\text{C=Ru})$, with a second-order perturbation energy ($E^{(2)}$) of 7.6 kcal/mol. $E^{(2)}$ represents the energy contribution that stabilizes the system. This indicates that the hyperconjugative effect of the C-B bond significantly stabilizes the C=Ru bond, and this stabilizing effect contributes substantially to the reduction of the energy of **INT-2 β -anti**. In contrast, no such orbital interaction is observed in **INT-2 γ -anti**, which exhibits higher energy. Therefore, we infer that the hyperconjugative interaction between $\sigma(\text{C-B})$ and $\pi^*(\text{C=Ru})$ likely plays a crucial role in driving the β -regioselectivity during the concerted hydrometallation process.

12. In Scheme 6A, **INT-2 β -anti** undergoes isomerization to **INT-2 β' -anti**, followed by rotation of the C β -C γ bond. The isomerization from **INT-2 β -anti** to **INT-2 β' -anti** involves an exchange of positions between the L ligand and the $-\text{Si}(\text{OEt})_3$ group, which seems improbable. The authors are requested to provide detailed computational results of the isomerization process.

Response:

Based on your suggestion, the isomerization process from **INT-2 β -anti** to **INT-2 β' -anti** is considered. First, the MeCN ligand in **INT-2 β -anti** dissociates from the Ru center to form **INT-noLigand**. Then, MeCN recoordinates (**INT-noLigand**) to form **INT-2 β' -anti**. The intermediate **INT-noLigand** has an energy of $\Delta G = 6.7 \text{ kcal/mol}$ relative to **INT-2 β -anti** ($\Delta G = 2.5 \text{ kcal/mol}$), indicating that the process occurs readily.

13. In Scheme 6A, the authors are requested to present the mechanistic process of direct C β -C γ bond rotation from **INT-2 β -anti**. This represents the most straightforward mechanism, so why was it not considered by the authors? The authors should explain the necessity of **INT-2 β' -anti** as an intermediate in the mechanism.

Response:

Thank you for your valuable suggestion. In our previous study, we have already attempted to locate the transition state for the direct rotating C β -C γ bond from **INT-2 β -anti**. In **INT-2 β -anti**, the Si(OEt)₃ group is positioned near the butyl chain side. The dihedral angle scan results reveal that, during the rotation of the C β -C γ bond, the hydrogen atom exhibits a stronger tendency to form an H-Si bond with the Si(OEt)₃ group, which consequently drives the reaction toward retro-hydrometallation. Given these challenges, and considering that the isomerization from **INT-2 β -anti** to **INT-2 β' -anti** is facile, we proceeded from the isomer **INT-2 β' -anti** and successfully obtained the transition state **TS β -rot** for C β -C γ bond rotation (seeing the figure below). Based on your suggestion, we have added an explanation in the manuscript as to why direct rotation of the C β -C γ bond is not feasible.

14. What does the double wavy line between TS-2β-anti and INT-3β-anti represent? If it means that some steps have been omitted between the two, please give details of the steps in the supporting material.

Response:

We are sorry for making it unclear to you. No steps are omitted here. This step is highly exothermic, and it is difficult to represent it proportionally to the energy levels in the diagram, so wavy lines are used instead. To avoid any misunderstanding, we changed the double tilde to a double hyphen.

15. Similar to comment 11, for Scheme 6B, the authors claim that the β-regioselectivity is attributed to the hyperconjugative effect between $\sigma(\text{C-B})$ and $\pi^*(\text{C}=\text{Ru})$. To assess the rationality of this inference, it is imperative to examine CpTS-1γ and CpTS-1β, rather than solely focusing on the non-selectivity-determining intermediates CpINT-2β-syn and CpINT-2γ-anti.

Response:

We appreciate your thoughtful consideration. Similar to the response in comment 11, according to the Hammond postulate (*J. Am. Chem. Soc.* **1955**, *77*, 2, 334–338), in endothermic reactions, the structures of the transition state tend to more closely resemble the products, and the greater the stability of the products, the lower the energy of the transition state.

Hammond postulate

Since the energy difference between the products **INT-2 β -anti** and **INT-2 γ -anti** is more significant and follows a similar trend to that of the transition states **TS-1 β** and **TS-1 γ** , analyzing the key three-membered ring intermediates **INT-2 β -anti** and **INT-2 γ -anti** can provide valuable insight into why **TS-1 β** is more stable.

16. The authors should attempt to rotate the H-Si(OEt)₃ group in CpINT-1 β to form an intermediate where the hydrogen is oriented inwards. Furthermore, they should investigate the possibility of obtaining CpINT-2 β -anti from this intermediate.

Response:

Thank you for your thoughtful suggestion. Prior to submitting our manuscript, we made numerous attempts, but unfortunately, we were unable to locate the transition state directly from ^cPINT-1 β to ^cPINT-2 β -anti. Therefore, we conducted BOMD calculations from ^cPTS-1 β .

In fact, in ^cPTS-1 β , the C _{γ} -C _{β} -Ru-H moiety is nearly planar, with a dihedral angle of -5.2 degrees, and the hydrogen atom is slightly inward (toward the Cp ring), which corresponds to "the hydrogen is oriented inwards" you mentioned above.

BOMD (Born-Oppenheimer Molecular Dynamics) results show that starting from ^cPTS-1 β , when the hydrogen atom transfers to C _{γ} , once the C _{γ} -H bond forms, the C _{β} -C _{γ} bond begins to rotate. We observed the existence of a metastable planar σ -vinyl intermediate, which has a lifetime of approximately 1 ps before converting into either ^cPINT-2 β -anti or ^cPINT-2 β -syn. When the hydrogen atom rotates upward (toward the Cp ring), ^cPINT-2 β -syn is generated; when H rotates downward (away from the Cp ring), ^cPINT-2 β -anti is produced. However, since ^cPINT-2 β -syn is thermodynamically more stable than ^cPINT-2 β -anti, the majority of trajectories starting from ^cPTS-1 β lead to the formation of ^cPINT-2 β -

syn (please see the figure below). This observation aligns with previously reported studies (*J. Am. Chem. Soc.* **2003**, *125*, 38, 11578; *Angew. Chem. Int. Ed.* **2015**, *54*, 5632; *J. Am. Chem. Soc.* **2020**, *142*, 13867).

A. Sum of the BOMD Results

B. Conformational Analysis of the Dynamic Trajectories

17. Based on the experimental results shown in entry 5 of Table 1 (where the MeCN ligand is replaced with PPh₃ in the presence of Cp, leading to an increase in E-selectivity) and entry 6 (where the ligand is changed to PPh₃ in the presence of Cp*, resulting in an improvement in Z-selectivity), it is evident that, apart from Cp and Cp*, the ligands MeCN and PPh₃ also influence stereoselectivity. The authors are requested to analyze and explain the impact of the differences between MeCN and PPh₃ on stereoselectivity, an important aspect that is not addressed at all in the paper.

Response:

When 5 mol% of AgPF₆ was added to precipitate the chloride from the catalyst, we observed a complete loss of stereoselectivity. When 10 mol% of PPh₃ was added to the reaction with [CpRu(MeCN)₃]PF₆ as the catalyst, the reaction did not occur. When both PPh₃ and LiCl were added to this catalyst system, good regio- and stereoselectivity were observed. All these results above indicated the crucial role of the Cl ligand in determining the stereochemical outcome of the reaction instead of PPh₃ (Scheme 5d).

d) the ligand effect

Based on additional DFT calculations, we found that in the $\text{CpRu(PPh}_3)_2\text{Cl}$ catalyst, the dissociation of the Cl ligand from the Ru metal requires a significantly higher energy of 48.2 kcal/mol, whereas the departure of the PPh_3 ligand from Ru is much more facile, with an energy of only 11.3 kcal/mol (see the Figure below). Therefore, regarding the catalysts used in Table 1, it is reasonable to assume that for both $\text{CpRu(PPh}_3)_2\text{Cl}$ or $\text{Cp}^*\text{Ru(PPh}_3)_2\text{Cl}$, the two PPh_3 ligands are ultimately replaced by the added alkyne and silane (as seen in **C^pINT-1 β**). It is the Cl ligand, rather than the PPh_3 ligands, that plays the pivotal catalytic role. This was further supported by a control experiment in which the addition of 5 mol% AgPF_6 to precipitate the chloride from the $\text{CpRu(PPh}_3)_2\text{Cl}$ catalyst resulted in a complete loss of stereoselectivity, underscoring the critical role of the Cl ligand in determining the stereochemical outcome of the reaction. And as for the hexafluorophosphate-type catalysts $[\text{CpRu(MeCN)}_3]\text{PF}_6$ or $[\text{Cp}^*\text{Ru(MeCN)}_3]\text{PF}_6$, two MeCN ligands are displaced by the added alkyne and silane, leaving one MeCN ligand coordinated to the Ru metal (as seen in **INT-1 β**).

Dissociation energy of the ligand in $\text{CpRu(PPh}_3)_2\text{Cl}$

Control experiment

The added alkyne and silane displace the ligands

Therefore, based on the actual ligand conditions in the aforementioned catalyst system, the mentioned "impact of the differences between MeCN and PPh₃ on stereoselectivity" should in fact be corrected to the differences in stereoselectivity between MeCN and Cl ligands. By analyzing the BOMD results, we can infer the underlying reasons for the changes in E or Z stereoselectivities in Table 1.

According to the BOMD calculation results, when the Cp* ring is combined with the MeCN ligand (**TS-1 β**), downward movement of the hydrogen atom causes the CH₂-B(MIDA) group to move upward, closer to the Cp* ring, resulting in significant steric hindrance. This makes the formation of **INT-2 β -syn** less favorable. Conversely, upward movement of the hydrogen atom does not introduce such steric hindrance, making this pathway more favorable and leading to the formation of the *anti*-product precursor (Table 1, entry 2, with [Cp*Ru(MeCN)₃]PF₆).

In the case where the Cp* ring is replaced with the Cp ring (Cp ring combined with the MeCN ligand), when the H atom moves downward, it is inferred that there is no steric hindrance due to the smaller size of the Cp ring compared to Cp*. This will increase the favorability of the pathway generating the *syn*-product precursor, leading to an increase in E-selectivity. This reasoning is supported by the experiment (Table 1, entry 4, with [CpRu(MeCN)₃]PF₆).

When the smaller Cp ring is combined with the smaller Cl ligand (**^{Cp}TS-1 β**), neither upward nor downward movement of the H atom introduces significant steric hindrance. However, since **^{Cp}INT-2 β -syn** is thermodynamically more stable than **^{Cp}INT-2 β -anti**, the majority of trajectories starting from **^{Cp}TS-1 β** generate *syn*-product precursor (Table 1, entry 5 or 7, with CpRu(PPh₃)₂Cl).

In the case where the Cp ring is replaced with the Cp* ring (Cp* ring combined with

the smaller Cl ligand), when the hydrogen atom moves downward, we inferred that the steric hindrance increases due to the larger size of the Cp* ring compared to Cp. This will reduce the favorability of the pathway generating the *syn*-product precursor, leading to an increase in Z-selectivity. This reasoning is supported by the experiment (Table 1, entry 6, with Cp*Ru(PPh₃)₂Cl).

Reviewer #3 (Remarks to the Author):

The manuscript by Wang, Sun, and coworkers describes a Ru-catalyzed regio- and stereoselective hydrosilylation of propargylic B(MIDA). Transition metal-catalyzed hydrosilylation of alkynes offers a direct and atom-economical approach to synthesizing alkenylsilanes, a family of useful compounds. However, current state-of-the-art methods primarily focus on the hydrosilylation of terminal alkynes. The use of unsymmetrical internal alkynes has faced challenges in achieving precise regio- and stereocontrol. Here, the authors achieve regioselective hydrosilylation of internal alkynes under Ru catalysis by harnessing the β-boron effect. Stereodivergent control is also realized by subtle variations in the Ru catalyst, which enables the synthesis of diverse regio- and stereo-defined building blocks that amalgamate the synthetic potential of boryl, silyl, and alkene functionalities. Control experiments and DFT studies were conducted to shed light on the detailed mechanism. Overall, I support the acceptance of the manuscript if the following points are sufficiently addressed.

1. The content in the third paragraph of the Introduction repeats information already presented in the second paragraph.

Response: Thanks. We have deleted the repeating paragraph.

2. For the *syn*-hydrosilylation reaction, CpRu(PPh₃)₂Cl has been identified as the optimal catalyst. Does the presence of PPh₃ have any significant impact on the reaction? It would be beneficial to conduct control experiments to know more about this point.

Response:

Thank you for your suggestion. Based on additional DFT calculations, we found that in the CpRu(PPh₃)₂Cl catalyst, the dissociation of the Cl ligand from the Ru metal requires a significantly higher energy of 48.2 kcal/mol, whereas the departure of the PPh₃ ligand from Ru is much more facile, with an energy of only 11.3 kcal/mol. Therefore, it is reasonable to assume that the two PPh₃ ligands are ultimately replaced by the added alkyne and silane (as seen in **CpINT-1β**) and thus the Cl ligand plays the pivotal catalytic role rather than the PPh₃ ligands.

This was further supported by a control experiment in which the addition of 5 mol% AgPF₆ to precipitate the chloride from the CpRu(PPh₃)₂Cl catalyst resulted in a complete loss of stereo-control, underscoring the critical role of the Cl ligand in determining the stereochemical outcome of the reaction.

Dissociation energy of the ligand

The added alkyne and silane displace two PPh₃ ligands

Control experiment

3. Regarding the substrate scope, could aromatic alkynes and enynes be applicable to this reaction? Exploring these substrates would help broaden the scope and applicability of the reaction.

Response:

Thank you for your advice. We have attempted the enyne **S-38** and aromatic alkyne **S-39**. **S-38** is applicable when using [Cp**Ru*(MeCN)₃]PF₆ as the catalyst to afford **44** in 76% yield with both good regio- and stereoselectivity but not applicable when using CpRu(PPh₃)₂Cl as the catalyst. Unfortunately, **S-39** is not applicable in both two Ru catalytic system.

(Z)-6-methyl-2-(2-(triethoxysilyl)penta-2,4-dien-1-yl)-1,3,6,2-dioxazaborocane-4,8-dione (44)

¹H NMR (500 MHz, Acetone-*d*₆) δ 6.87 – 6.79 (m, 1H), 6.72 (d, *J* = 11.2 Hz, 1H), 5.15 – 5.03 (m, 2H), 4.16 (d, *J* = 16.7 Hz, 2H), 3.96 (d, *J* = 16.7 Hz, 2H), 3.82 (q, *J* = 7.0 Hz, 6H), 3.10 (s, 3H), 1.77 (s, 2H), 1.17 (t, *J* = 7.0 Hz, 9H).

¹³C NMR (126 MHz, Acetone-*d*₆) δ 168.6, 145.7, 138.1, 136.4, 117.0, 62.5, 58.8, 46.0, 18.5.

These results were included in the manuscript.

4. In Scheme 3, the labeling of the products in the model reaction, 'Z isomer' should be 'E isomer'. Please check.

Response: Thank you for your advice. Corrected.

5. It would be valuable to examine a gram-scale experiment to evaluate the robustness and practicality of the reaction on a larger scale.

Response:

Thank you for your advice. As shown in Scheme 4, the gram scale synthesis of **6** was conducted to provide the product in 87% yield after direct crystallization.

6. In the DFT calculation section, more transition states should be considered. Specifically, the transition state involving planar σ -vinyl intermediates should be included, particularly from TS-1 β to INT-2 β -anti. In this case, two different ruthenacyclopentene intermediates can be formed through clockwise or counterclockwise rotation of the σ -vinyl ruthenium. Similarly, calculations for the reaction involving syn-selectivity should also be conducted.

Response: Thank you for your suggestion. Prior to submitting our manuscript, we made multiple attempts to obtain the planar σ -vinyl intermediate, but unfortunately, all optimization attempts ultimately led to the formation of **INT-2 β -anti**. In the BOMD (Born-Oppenheimer Molecular Dynamics) trajectories starting from **TS-1 β** (among 100 dynamic trajectories, 58 led to **INT-1 β** preceding **TS-1 β** , 42 resulted in **INT-2 β -anti** and **INT-2 β -syn**), we observed the existence of a metastable planar σ -vinyl intermediate, which has a lifetime of approximately 1 ps before converting into either **INT-2 β -anti** or **INT-2 β -syn**. As shown in the Figure below, when the hydrogen atom moves downward, the CH₂-B(MIDA) group moves upward, approaching the Cp* ring, resulting in significant steric hindrance, which disfavors the formation of **INT-2 β -syn** (among 100 dynamic trajectories, only 1 trajectory yielded **INT-2 β -syn**). In contrast, in the dynamic trajectories where the hydrogen atom moves upward, no significant steric hindrance is introduced, making this pathway more favorable and leading to the formation of **INT-2 β -anti** (41 out of 100 trajectories produced **INT-2 β -anti**). Overall, based on our attempts and the analysis of BOMD results, the planar σ -vinyl intermediate is unstable.

7. To further validate the computational results, the authors are encouraged to perform additional mechanistic experiments, such as deuteration experiments, kinetic isotope effect (KIE) studies, and kinetic studies.

Response:

Thank you for your advice.

Deuteration experiments of **S-1** with DSiEt₃ (99% D) yield **3-d** and **46-d** smoothly in 99% deuteration level at the γ -position with both two Ru catalytic system.

Kinetic isotope effect measured via intermolecular competition between HSiEt₃ and DSiEt₃ gave an inverse KIE ($k_H/k_D = 0.67$, $k_H/k_D = 0.47$) with those Ru catalysts. Similar to the result above, a parallel kinetic isotope experiment with CpRu(PPh₃)₂Cl as the catalyst gave an inverse KIE of 0.32. We proposed that during the concerted oxidative-addition and hydride-insertion, the weaker Si-H/D bond are partially weakened and C-H/D bond are partially formed. Due to the higher bond energy of C-D bond than C-H bond, the driving force of forming C-D bond should be stronger. This phenomenon was also observed in several previous studies (Organometallics 2024, 43, 3236; Chem. Eur. J. 2009, 15, 11515; Chem. Rev. 2011, 111, 4857-4963).

e) intermolecular KIE experiment

f) parallel KIE experiment

For SI:

1. The mass of the products should be included in the SI, rather than just providing the yield.

Response: Thank you for your advice. All the mass of the products has been shown in SI.

2. Page 43 in the SI, product 47, 'calculated for ...Si...' should be 'calculated for ...Si3(subscript)...'.

Response: Thank you for your advice. The number '3' has been subscripted.

ESI-MS: calculated for $C_{19}H_{40}BNNaO_6Si_3^+$ $[M+Na]^+$, 496.2149; Found, 496.2146.

3. Page 43 in the SI, product 48, there is no methyl group information in the 1H NMR data.

Response: Thank you for your advice. The methyl group information '0.90 (t, $J = 7.0$ Hz, 3H)' has been added.

4. Page 174 in the SI, for product 24, the chemical shifts in the copy of 1H NMR spectrum should be masked, and the chemical shifts in the 13C NMR spectrum are inconsistent with the corresponding data on page 31.

Response:

Thank you for your advice. Corrected.